# Intra and Inter-Test Reliability of Isometric Hip Adduction Strength Test with Force Plates in Professional Soccer Players

**DOI:** 10.3390/jfmk9040270

**Published:** 2024-12-12

**Authors:** Jorge Pérez-Contreras, Juan Francisco Loro-Ferrer, Pablo Merino-Muñoz, Felipe Hermosilla-Palma, Brayan Miranda-Lorca, Alejandro Bustamante-Garrido, Felipe Inostroza-Ríos, Ciro José Brito, Esteban Aedo-Muñoz

**Affiliations:** 1Escuela de Doctorado de La Universidad de Las Palmas de Gran Canaria (EDULPGC), 35016 Las Palmas, Spain; jperez51@santotomas.cl; 2Escuela de Ciencias del Deporte y Actividad Física, Facultad de Salud, Universidad Santo Tomas, Santiago 8370003, Chile; alejandro.bustamante@umce.cl; 3Clinical Sciences Department, University of Las Palmas de Gran Canaria, 35016 Las Palmas, Spain; juanfrancisco.loro@ulpgc.es; 4Núcleo de Investigación en Ciencias de la Motricidad Humana, Universidad Adventista de Chile, Chillán 3780000, Chile; pablo.merino@peb.ufrj.br; 5Biomedical Engineering Program, Federal University of Rio de Janeiro, Rio de Janeiro 21941-617, Brazil; 6Escuela de Pedagogía en Educación Física, Facultad de Educación, Universidad Autónoma de Chile, Talca 3460000, Chile; felipe.hermosilla@uautonoma.cl; 7Departamento de Educación Física, Deportes y Recreación, Facultad de Artes y Educación Física, Universidad Metropolitana de Ciencias de la Educación, Santiago 7760197, Chile; brayan.mirada2019@umce.cl; 8Department of Physical Education, Federal University of Juiz de Fora, Governador Valadares 35010-180, Brazil; felipe.inostroza.311@gmail.com (F.I.-R.); ciro.brito@ufjf.br (C.J.B.); 9Escuela de Ciencias de la Actividad Física, El Deporte y la Salud, Facultad de Ciencias Médicas, Universidad de Santiago de Chile, Santiago 8370003, Chile

**Keywords:** reproducibility, muscle strength, isometric contraction, hip

## Abstract

Assessing the reliability of measurement instruments and equipment is essential to ensure the accurate tracking of athletes over extended periods, minimizing the measurement errors caused by chance or other factors. However, a less common but equally important analysis is the verification of inter-measurement agreement, which complements the reliability results. **Purpose:** To evaluate the intra- and inter-test reliability of an isometric hip adduction strength and asymmetries test in professional soccer players. **Methods**: Twenty-three professional male soccer players were evaluated on two occasions, 1 week apart. The force signal was collected using force plates (Pasco PS-2141), and the data processing was performed using Matlab software (R2023a, MathWorks, Natick, MA, USA). The peak force, interval RFD, peak RFD, peak force asymmetry and RFD were analyzed. Intraclass correlation coefficients (ICCs) and coefficients of variation (CV) were calculated to corroborate the intra- and inter-test reliability. In addition, the degree of agreement of the asymmetries was corroborated through the kappa index. **Results**: The peak force demonstrated an acceptable absolute reliability (CV < 10%) for the intra-test and test–retest condition, an excellent relative intra-test reliability and a good to moderate reliability for the test–retest reliability. However, the peak force asymmetry showed a moderate test–retest reliability and agreement. For the intra-test condition, the RFD variables demonstrate a moderate to excellent relative reliability; however, all demonstrate unacceptable absolute reliability (CV > 10%) in at least one of the evaluation sessions. A moderate to poor test–retest reliability and unacceptable absolute reliability were observed for all the RFD variables. **Conclusions**: The peak force is the variable with the highest intra- and inter-test reliability, so its use is recommended to longitudinally assess the maximum strength of the adductors in professional soccer players, but not the asymmetry orientation of the peak force. The RFD variables should be interpreted with caution due to their inconsistent reliability, and it is necessary to improve the methods used to achieve adequate reliability.

## 1. Introduction

The incidence of injuries associated with the practice of physical activity is common in both amateur and professional athletes and has been described in team and individual sports [1,2,3]. An injury can lead athletes to the loss of training and competitions, and in reference to soccer, the costs associated with an injury can reach 500,000 euros in a European league [4], so, decreasing the risk of injury becomes crucial for sports clubs. In soccer, according to Raya-González and Estevez Rodríguez [5], there are extrinsic and intrinsic risk factors that can increase the probability of suffering an injury; among these factors, they describe some, such as the playing surface, time of the season and muscle strength. It has been shown that the lower limbs are the area where soccer players suffer the most injuries, with an average rate of 6.8 injuries out of a total of 8.1 per 1000 h of training and competitive participation [6]. Within this segment, the adductor musculature is one of the most affected [7], and the weakness of the adductor musculature and abnormal muscle ratios showing asymmetries (differences between the legs) have been shown to be risk factors for new inguinal injuries, such as Adductor Strain or Osteitis Pubis [8]. Adductor strains are usually located in the region of the myotendinous junction, occurring during actions such as kicking, sprinting and changes of direction [9]. On the other hand, Osteitis Pubis is an inflammatory lesion in the pubic symphysis due to traumatic stresses or repeated efforts, usually associated with a mismatch between the strength of the adductor and abdominal musculature [10]. Also, the percentage of asymmetries between limbs has been a widely studied indicator, finding that a high asymmetry between the limbs is moderately associated with the risk of suffering indirect injuries in the lower limb [11], in addition to affecting the performance of actions such as jumps and changes in direction [12]. Therefore, it is particularly important to assess the different dimensions of strength for both injury prevention and physical performance.

To fulfill this purpose, maximal isometric strength measurements are widely used, being able to assess the maximal force production as well as the rate of force development (RFD) [13]. Previously, adductor isometric strength has been assessed through hand-held dynamometers demonstrating good intra-test, inter-test and inter-rater reliability values for the maximal strength variables [14,15,16]. To our knowledge, adductor testing has not been performed with force platforms, which are instruments that already have portable options [17,18] and with which several dynamic and isometric tests can be performed, including isometric mid-thigh pull, jumping with countermovement, and isometric hamstring tests at 30° and 90° of knee flexion [19,20,21]. Therefore, the evaluation of adductor strength through force platforms presents a viable option in terms of data collection of the force-time curve, lower cost of equipment and a suitable implementation for the environments in which soccer players perform.

Assessing the reliability of measurement instruments and equipment is essential to ensure the accurate tracking of athletes over extended periods and minimizing the measurement error caused by chance or other factors [22,23]. However, a less common, but equally important analysis is the verification of inter-measurement agreement, which complements the reliability results. The between-measurement agreement complements the reliability results. Agreement assesses the similarity between mediations performed by different assessors or at different times, and a high degree of agreement facilitates the detection of substantial changes between measurements over time [24]. Therefore, the aim of this study was to evaluate the intra- and inter-test reliability of an isometric strength test of the hip adductors in professional soccer players, in order to establish its usefulness as a monitoring tool in performance and injury prevention.

## 2. Materials and Methods

### 2.1. Design

A quantitative, non-experimental, descriptive study with a cross-sectional design was used [25].

### 2.2. Sample

The study sample consisted of 23 male professional soccer players (age = 21.6 ± 1.4 years; weight = 70.3 ± 6.7 kg; height = 176 ± 6.5 cm), belonging to a professional second division club in Chile. The sample was selected via convenience sampling. The inclusion criteria for participation in the study were to (i) complete at least three valid attempts for each leg in the test performed, (ii) not having suffered inguinal or adductor muscle injuries in the last 6 months and (iii) have at least 3 years of professional soccer experience. The exclusion criterion was presenting discomfort in the lower limbs during the test.

Before initiating the study, the club was initially contacted through e-mails addressed to the technical directors and physical preparation coordinators, explaining the objectives of the study, the procedures involved and the potential benefits of participating. Once the club approved the collaboration, the athletes were invited through informative meetings held at their facilities. Subsequently, the participants were informed about the study objectives, procedures and associated risks through an online form, which they accessed from their mobile devices, complying with the standards established in the Declaration of Helsinki [26].

### 2.3. Procedures

The players participated in an initial familiarization session. The evaluation was carried out on the following day, corresponding to the second day of a preseason microcycle, and the second evaluation was carried out 7 days later. All the evaluations were carried out at the club’s facilities, in the morning and before the technical–tactical practice.

For the characterization of the sample, an ISAK Level II certified professional recorded the height of the participants using a Seca 216 mechanical wall stadiometer. The body weight was measured using two force plates (Pasco PS-2141, Roseville, CA, USA). The athletes were instructed to adopt a standardized posture with hands on hips and gaze straight ahead, holding this position for 2 s. The data recorded during this time were averaged to determine the body weight in Newtons (N), and subsequently divided by the acceleration of gravity to convert it to kilograms (kg). Two evaluators with experience in evaluations and data acquisition on force platforms recorded the signals during the physical evaluations.

### 2.4. Data Recording

Prior to the data recording, the participants underwent a warm-up structured in three blocks:

Low-intensity continuous running: This consisted of light jogging with a subjective perception of effort of 3, accompanied by joint mobility exercises and static and dynamic stretching, with a focus on the lower body, especially the adductor muscles of the hip. This block lasted 10 min.

Specific lower-body self-loading exercises: Three sets of 3 repetitions were performed on a coordination rail, followed by 2 sets of 10 repetitions of squats. Also included were 2 sets of 3 repetitions of forward lunges with a 2 s pause in the maximum flexion position. The perceived exertion in this block was not to exceed a 4 on the subjective scale.

Strength activation exercises: This block included specific exercises for the lower body that incorporated isometric and dynamic contractions, optimizing muscle activation prior to the test. Three 5 s sets of maximal isometric contractions were performed on the hip adductors with a medicine ball (between the knees), followed by 3 sets of 8 repetitions of lateral strides for each leg. In this block, the perceived exertion was not to exceed a 5 on the subjective scale.

#### Isometric Hip Adduction Strength Test

Pasco uniaxial force plates (PS-2141, Roseville, CA, USA) and a high-density foam mat (3 cm thick) were used for the test. The protocol proposed by Lovell [27] was followed. For the data acquisition, the participants were required to adopt a lateral stance position, ensuring full contact of the medial edge of the foot on a force plate. The hands were placed on the shoulders contralaterally. The tested leg was to be held with the hip and knee fully extended at a 0° angle, while the opposite leg was positioned with the hip and knee flexed at 90° (see Figure 1). Before starting the test, the athletes were instructed to perform two preliminary attempts per leg, with the perceived exertion levels between 5 and 8 on a scale of 1 to 10 (where 10 indicates maximum effort). This procedure was intended to verify the correct adoption of the test position and to ensure participant comfort. Subsequently, each participant made 3 attempts of a 3 s duration, with a recovery interval of 30 s between each attempt. The cue from the evaluators was: “push the floor as fast and hard as possible” [28]. The recording of the signals began with a countdown “3,2,1, go”. Upon announcing “go”, the participants exerted their maximum possible force, while being verbally encouraged by the evaluators with the command “push”. The two attempts that achieved the highest peak force values were selected for the reliability analysis.

### 2.5. Data Processing

Signals were acquired using the CAPSTONE software version 2.2.2 (Pasco, Roseville, CA, USA) with a sampling rate of 1000 Hertz and imported into a spreadsheet. They were then imported into the MATLAB^®^ software (R2023a, MathWorks, Natick, MA, USA) for processing by means of a script made by the authors. First, the signals were filtered by a 10 Hertz low-pass filter. Then, the onset of the signal was determined by using the 5 standard deviations method. The following variables were calculated: the peak force, force development rate from 0 to 50, 0 to 100 ms and from 100 to 200 ms and peak force development rate through a 20-sample moving average with a one-sample overlap. The force and force development rate signals can be seen in Figure 2.

### 2.6. Statistical Analysis

First, the normality of the data distribution was assessed using the Shapiro–Wilk test. To measure the intra-test and inter-test reliability on an absolute scale, the coefficient of variation (CV) was used. The CV was calculated for each athlete and then the sample average was obtained. Absolute values of less than 10% were considered acceptable according to the criteria established by Atkinson and Nevill [29]. To assess the relative reliability, the intra class correlation coefficient (ICC) was used using a two-factor mixed effects model [30]. The ICC values were categorized according to the thresholds defined by Koo and Li [31]: values below 0.49 were considered poor; 0.5 to 0.74 were classified as moderate; 0.75 to 0.89 were considered good; and values above 0.9 were classified as excellent. To evaluate the degree of absolute agreement in individual measurements, Cohen’s Kappa index was used, which was interpreted according to the following categories: <0.00 no agreement, 0.00–0.20 negligible agreement, 0.21–0.40 medium agreement, 0.41–0.60 moderate agreement, 0.61–0.80 substantial agreement and 0.81–1.00 near perfect agreement [32]. The 95% confidence intervals (CI) were also assessed. Statistical analyses were performed using SPSS version 25, and the significance level was set at an alpha of 0.05. In addition, Bland–Altman plots were generated using GraphPad Prism 8 to further examine the residual scores [33].

## 3. Results

Table 1 shows excellent reliability in the peak force variables (CV: R = 2% and L = 2%; ICC: R = 0.97 and L = 0.98), RFD200 (CV: R = 18% and L = 17%; ICC: R = 0.98 and L = 0.93), the peak RFD (CV: R = 7% and L = 9.6%; ICC: R = 0.94 and L = 0.94) and also in the RFD100 variable in the right leg (CV: R = 18%; ICC: R = 0.94). Good absolute reliability is shown in RFD50 (CV: R = 39% and L = 38%; ICC: R = 0.81 and L = 0.9) and in RFD100 in the left leg (CV: L = 25%; ICC: L = 0.82).

Table 2 shows excellent absolute reliability in the peak force variables (CV: R = 2% and L = 3.6%; ICC: R = 0.97 and L = 0.94) and the peak RFD in the right leg (CV: R = 14.6%; ICC: R = 0.95). Good absolute reliability is shown in RFD100 (CV: R = 26.5% and L = 25%; ICC: R = 0.81 and L = 0.83) and RFD200 in the right leg (CV: R = 27.8%; ICC: R = 0.83). Moderate reliability is shown in RFD50 (CV: R = 43% and L = 41%; ICC: R = 0.70 and L = 0.65) and RFD200 in the left leg (CV: L = 29%; ICC: L = 0.63) and in the peak RFD in the left leg (CV: L = 12.6%; ICC: L = 0.67).

Table 3 shows good absolute inter-test reliability in the variable peak force in the left leg (CV: L = 4.1%; ICC: L = 0.80), moderate in the peak force in the right leg (CV: R = 4.7%; ICC: R = 0.69), in RFD50 left leg (CV: L = 38%; ICC: L = 0.50), RFD100 left leg (CV: L = 22%; ICC: L = 0.66) and in the peak RFD (CV: R = 15% and L = 14%; ICC: R = 0.63 and L = 0.64), poor in the variables RFD50 right leg (CV: R = 44%; ICC: R = 0.18), RFD100 right (CV: R = 30%; ICC: R = 0.06), and in RFD200 (CV: R = 39% and L = 33%; ICC: R = −0.81 and L = −0.07). Figure 3 shows the Bland–Altman plots for the peak force of both limbs with their respective bias values in Table 4.

Table 5 shows the absolute reliability and level of agreement for the test–retest in the form of the percentage of bilateral asymmetry. A moderate absolute reliability and moderate peak agreement can be observed (CV = −42%; ICC = 0.52; k = 0.49). Figure 4 shows the descriptive values of the bilateral asymmetry of the peak force.

## 4. Discussion

The high incidence of injuries in sport, especially in soccer, highlights the need to identify the risk factors [6]. The lower limbs, particularly the adductor muscles, are the most affected, and their weakness is a key factor in the risk of groin injuries [4,8]. Assessing the reliability of measurement instruments is crucial for the accurate monitoring of athletes, reducing the errors caused by chance [22,23]. A less frequent but essential aspect is the analysis of agreement, which measures the similarity between assessments, either by using different evaluators or at different times, facilitating the detection of important changes over time [24]. The purpose of this study was to validate the internal and inter-test reliability of an adductor strength test in professional soccer players.

### 4.1. Intra-Test Reliability

Moderate to excellent relative reliability values were found for the variables analyzed during both evaluation sessions. The peak force presented the best values in regard to relative and absolute reliability considering both legs and both evaluation sessions, which is in agreement with the results of similar research evaluating the reliability in strength tests for adductors. A recent study (16) analyzed the compression strength and peak isometric torque in analysis using a manual dynamometer in a lateral decubitus position (hip and knee at 0°), obtaining excellent intra-day reliability results for both variables (ICC > 0.8; CV < 10%). For the RFD variables, moderate to excellent relative reliability values were obtained for the peak RFD and RFD200, as well as good to excellent values for RFD100 and moderate to good values for RFD50. However, RFD50, 100 and 200 exhibited an inadequate absolute reliability in their results, which is consistent with the low reliability exhibited by similar isometric assessments [34,35]. Specifically, an early RFD presents a higher variability which could be mainly associated with neural factors during the isometric assessment process [36]. Only the RFD peak achieved an acceptable value for absolute reliability, making it the most reliable RFD indicator to evaluate during a session.

These findings suggest that the isometric hip adduction strength test has adequate intra-test reliability and is an appropriate tool to measure the maximal strength and explosive strength in male professional soccer players, specifically represented by the peak force and peak RFD. However, the high dispersion presented by the RFD 50, 100 and 200 between attempts should be taken into account, emphasizing the need for adequate familiarization and intentionality on the part of the athlete to decrease the variation in the results [36].

### 4.2. Reliability and Agreement Test–Retest

The results of the inter-test analysis identify the peak force and peak RFD as the most reliable variables, with good to moderate relative reliability for the peak force and only moderate for the peak RFD. In the studies by Denton [14] and Kirwan [16] the reliability of the peak isometric torque peak using a manual dynamometer in a lateral decubitus position (hip and knee at 0°) was analyzed, obtaining excellent results (ICC = 0. 94 and SEM = 7%; ICC = 0.99 to 0.99 and CV = 1.2 to 1.42%, respectively). In addition, Kirwan [16] evaluated the reliability of the adductor compression force, also with excellent reliability (ICC = 0.99 to 0.99; CV = 2.2 to 4%). A similar assessment, but using isokinetic dynamometry assessed the torque during a maximal voluntary contraction, also with excellent absolute and relative reliability (ICC = 0.90 to 0.98; SEM = 5.1 to 10.1%, Bias = −2.9 to 6%) [37]. Although the metrics analyzed, equipment used and evaluation position are not the same, they result in a useful comparison in the inter-test reliability testing of this research. For absolute reliability, the peak force presented acceptable values (CV < 10%) for the dispersion of the results between the first and second measurement sessions. On the other hand, the Bland–Altman analysis identified a low bias for the peak force of the right and left leg, which can be interpreted as a high degree of agreement of the results between both sessions. Therefore, it appears that the peak force proves to be the most consistent variable in terms of the absolute and relative test–retest reliability of the isometric hip adduction strength test. Its adequate absolute reliability will allow professionals in the area to detect substantial changes over time, decreasing the possibility that improvements are misinterpreted by the variable’s measurement error [24].

Like the intra-test reliability analysis, the absolute and relative reliability results for the RFD variables are unfavorable, which is again in line with the previous findings of a low RFD reliability in mono- and multi-joint assessments [36], whose alterations could be explained by physiological or methodological factors [38]. Therefore, a better systematization of the data collection and analysis procedures is required, as well as considering possible confounding factors that alter the reliability between tests, to ensure adequate replicability of the results and an adequate assessment of the RFD through the isometric hip adduction strength test.

In terms of the percentage of bilateral asymmetry of the peak force, moderate reliability and agreement were observed. The latter can be corroborated by visual inspection of Figure 3, where 17 players maintained the direction of the asymmetry and 6 players reversed the orientation of the asymmetry. In reviewing the literature, it seems to be an unusual practice to analyze the inter-test reliability and agreement of bilateral symmetry. Perez-Castilla [39] found unacceptable inter-test reliability for the peak force asymmetry for a bilateral (ICC = 0.15) and unilateral (ICC = 0.63) countermovement jump. On the other hand, the authors found adequate reliability (<0.7) for the mean force in the countermovement jump; this could be due to the fact that the peak force identified in the force–time curve only represents a discrete value and not necessarily the force production during the evaluation. Therefore, it could be more appropriate to analyze the mean force in future investigations. However, this should be noted in future research due to the differences between dynamic and isometric force manifestations. A recent study [40] analyzed the asymmetric orientation for a unilateral isometric mid-thigh pull, finding for the peak force a kappa index = 0.64 for a measurement between tests, which, in conjunction with our results, suggests that the peak force asymmetry is inconsistent over time when identifying the orientation of the asymmetry. Another interesting approach for asymmetry assessment is through Statistical Parametric Mapping to identify the asymmetries during the entire force–time curve, which may present more adequate inter-test reliability for this type of analysis [41]. These results highlight the need to deepen the reliability and agreement of bilateral asymmetry.

One of the limitations of the study lies in the sample size, since, although 23 professional soccer players were included, a larger number of participants could increase the robustness and generalization of the results to other sports populations [42].

Future studies could extend the sample to a more diverse and larger group, including both male and female soccer players of different competitive levels, which would allow us to explore the generalization of the results obtained. In addition, performing a longitudinal analysis over several seasons or training cycles would help to evaluate the evolution of adductor strength and the consistency of the measurements in different phases of training or competition. On the other hand, it would be relevant to investigate the relationship between adductor strength and the risk of groin or lower limb injuries, establishing whether the variations in measurements can predict or prevent future injuries. Finally, the incorporation of other biomechanical variables, such as electromyography measurements or the use of diagnostic imaging, could provide a more complete assessment of the adductor muscle function and its link to injury prevention [27].

The implementation of reliable adductor strength testing allows coaches and healthcare professionals to accurately and consistently assess the strength of this muscle group throughout the season, identifying weaknesses or asymmetries that facilitate the adjustment of training programs and improve performance. Since weakness in the adductors is a risk factor for groin injuries, this test can be integrated into regular monitoring protocols, helping to detect changes that increase the risk of injury and allowing early interventions. It also contributes to the optimization of physical performance by allowing the personalization of strength training and the reduction in muscle asymmetries.

Isometric hip adductor testing can be efficiently implemented in a team of 25 players, completing the assessment in approximately 45 min with two evaluators and two force plates, including the warm-up. Standard data analysis requires 4 to 5 h to produce a detailed report available the following day, although the use of premium licensing of the software allows key results, such as the peak force, to be obtained instantaneously, facilitating immediate feedback.

This methodology can be performed after warming up and before technical–tactical training, assessing the physical preparation and optimizing the daily load management. Furthermore, its practical implementation has proven to be compatible with the sports routine, allowing players to train without inconveniences one hour after the test.

## 5. Conclusions

In the isometric hip adduction strength test, the peak force is the variable with the highest intra- and inter-test reliability; therefore, its use is recommended to longitudinally assess the maximum strength of the adductors in professional soccer players. Due to the moderate reliability and agreement of the asymmetry of the peak force between the tests, it is suggested to develop other methods to identify the orientation of asymmetries in professional soccer players. The RFD variables should be interpreted with caution due to their inconsistent reliability, and it is necessary in the future to improve the methods used to achieve adequate reliability. The availability of a reliable test allows technical teams to make more informed decisions about the workload and the need for rehabilitation or adjustments in the training program, based on objective data on muscle strength.

## Figures and Tables

**Figure 1 jfmk-09-00270-f001:**
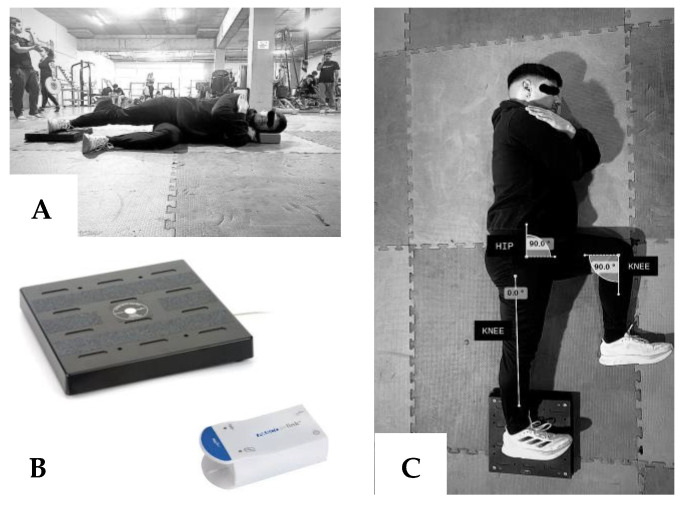
Isometric hip adduction strength test: side view (**A**), instrument (**B**) and top view (**C**).

**Figure 2 jfmk-09-00270-f002:**
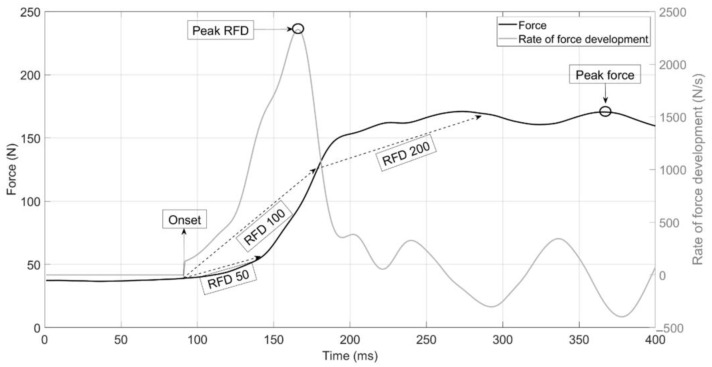
Signal and variables analyzed: peak force and rate of force development (RFD).

**Figure 3 jfmk-09-00270-f003:**
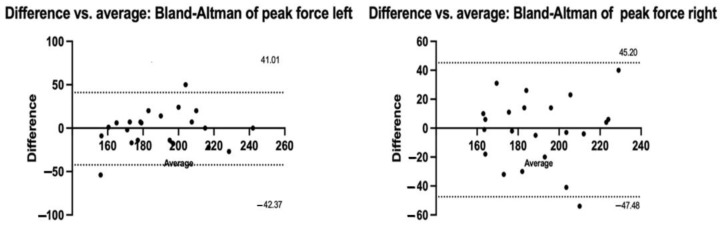
Bland–Altman plots of peak force in both profiles.

**Figure 4 jfmk-09-00270-f004:**
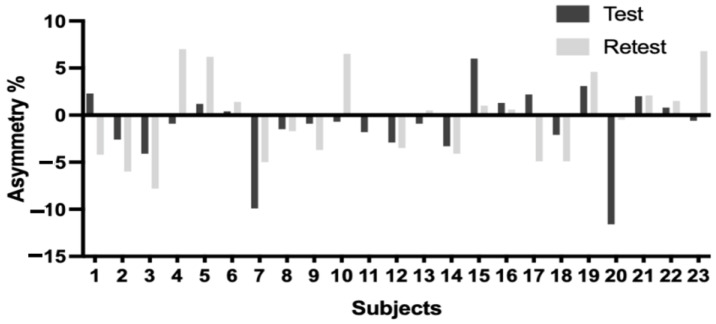
Peak adductor force asymmetries. Greater asymmetry in the non-dominant leg is represented by negative values, while positive values indicate greater asymmetry in the dominant leg.

**Table 1 jfmk-09-00270-t001:** Description and reliability of the adductor strength test time 1.

	LP	M	SD	CV	ICC	IL	UL	*p*	Reliability
Peak force (N)	R	192	4	2.0	0.97	0.94	0.99	0.001	Excellent
	L	190	3.7	2.0	0.98	0.97	0.99	0.001	Excellent
RFD50 (N/s)	R	581	202	39	0.81	0.55	0.91	0.001	Good
	L	597	173	38	0.90	0.76	0.95	0.001	Good
RFD100 (N/s)	R	824	90	18	0.94	0.86	0.97	0.001	Excellent
	L	715	134	25	0.82	0.59	0.92	0.001	Good
RFD200 (N/s)	R	821	75	18	0.98	0.95	0.99	0.001	Excellent
	L	781	105	17	0.93	0.84	0.97	0.001	Excellent
Peak RFD (N/s)	R	1748	134	7	0.94	0.86	0.97	0.001	Excellent
	L	1570	149	9.6	0.94	0.86	0.97	0.001	Excellent

LP limbs profile; M mean; SD standard deviation; ICC intra-class correlation coefficient; IL lower limit 95%; UL upper limit 95%; R right; L left; RFD50 rate force development 0 to 50 ms; RFD100 rate force development 0 to 100 ms; RFD200 rate force development 0 to 200 ms: Peak RFD peak rate of force development.

**Table 2 jfmk-09-00270-t002:** Description and reliability of the adductor strength test time 2.

	LP	M	SD	CV	ICC	IL	UL	*p*	Reliability
Peak force (N)	R	190	3.9	2.0	0.97	0.92	0.98	0.001	Excellent
	L	191	6.7	3.6	0.94	0.86	0.97	0.001	Excellent
RFD50 (N/s)	R	502	211	43	0.70	0.26	0.87	0.005	Moderate
	L	592.5	183	41	0.65	0.18	0.85	0.009	Moderate
RFD100 (N/s)	R	744.8	145	26.5	0.81	0.54	0.92	0.005	Good
	L	774	163	25	0.83	0.85	0.93	0.001	Good
RFD200 (N/s)	R	504	123	27.8	0.83	0.60	0.93	0.001	Good
	L	492	134	29	0.63	0.11	0.85	0.014	Moderate
Peak RFD (N/s)	R	1863	232	14.6	0.95	0.89	0.98	0.001	Excellent
	L	1669	228	12.6	0.67	0.19	0.86	0.008	Moderate

LP limbs profile; M mean; SD standard deviation; ICC intra-class correlation coefficient; IL lower limit 95%; UL upper limit 95%; R right; L left; RFD50 rate force development 0 to 50 ms; RFD100 rate force development 0 to 100 ms; RFD200 rate force development 0 to 200 ms: Peak RFD peak rate of force development.

**Table 3 jfmk-09-00270-t003:** Description and reliability test–retest of the adductor strength test.

	LP	M1	M2	SD1	SD2	CV	ICC	IL	UL	*p*	Reliability
Peak force (N)	R	190	191	25	24	4.7	0.69	0.25	0.87	0.005	Moderate
	L	190	190	28	25	4.1	0.80	0.52	0.92	0.001	Good
RFD50 (N/s)	R	528	458	432	344	44	0.18	−1.02	0.66	0.325	Poor
	L	502	600	385	477	38	0.50	−0.19	0.79	0.059	Moderate
RFD100 (N/s)	R	782	680	353	343	30	0.06	−1.26	0.61	0.443	Poor
	L	703	757	315	393	22	0.66	0.19	0.86	0.008	Moderate
RFD200 (N/s)	R	759	561	462	424	39	−0.81	−3.32	0.24	0.920	Poor
	L	771	478	404	225	33	−0.07	−0.90	0.47	0.590	Poor
Peak RFD (N/s)	R	1805	2129	588	1130	15	0.63	0.15	0.84	0.010	Moderate
	L	1561	1788	484	572	14	0.64	0.19	0.85	0.006	Moderate

LP limbs profile; M1 mean of first test; SD1 standard deviation of first test; M2 mean of second test; SD2 standard deviation of second test; ICC intra-class correlation coefficient; IL lower limit 95%; UL upper limit 95%; R right; L left.

**Table 4 jfmk-09-00270-t004:** Results of Bland–Altman statistics.

Peak De Force (N)	M Bias	SD Bias	LL	UL
Right	−1.14	23.64	−47.48	45.20
Left	−0.682	21.27	−42.37	41.01

M mean; SD standard deviation; IL lower limit 95%; UL upper limit 95%.

**Table 5 jfmk-09-00270-t005:** Description and reliability test–retest of the adductor strength % asymmetry.

	M1	M2	SD1	SD2	CV	ICC	IL	UL	*p*	Reliability	k	*p*	Agreement
Peak force	−1.1	−0.4	3.9	4.5	−42.6	0.52	0.14	0.80	0.04	Moderate	0.56	0.001	Moderate
Peak RFD	−7.3	−6.0	15.9	18.7	140	0.35	−0.59	0.73	0.16	Poor	0.45	0.001	Moderate

M mean; SD standard deviation; ICC intra-class correlation coefficient; IL lower limit 95%; UL upper limit 95%; R right; L left; K Cohen’s kappa.

## Data Availability

Data will be made available upon reasonable request.

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
