# Peer review of "Intra and Inter-Test Reliability of Isometric Hip Adduction Strength Test with Force Plates in Professional Soccer Players"

_jfmk, 2024, doi:10.3390/jfmk9040270_

Round 1
Reviewer 1 Report
Comments and Suggestions for Authors
This study is interesting because of its transferability to sports practice.
Introduction
The text should be marked with line numbers.
This study is very short and focuses mainly on descriptive statistical data. It would be more suitable to explain which injuries and in which specific movements they occur.
There are parts of the introduction that should not be in this section (from the 2nd paragraph onwards).
More citations on the use of platforms for this purpose are needed, and if not available, to look for sports with similar physiological demands.
Methods
Making it a requirement for inclusion that they can complete the test is a significant bias. Details such as experience, previous injuries, etc., which are not mentioned, should be considered.
Warm-up intensities are not quantitatively indicated, and this is an important aspect for the main part.
Discussion
It would be advisable to indicate why they have chosen a static action and not a dynamic one (with a dynamometer for this action). This would make it easier to do it in a standing position, as your sport is developed.
It could also be suggested to perform an eccentric test, which could also be quantified with inertial devices (at least these two considerations could be considered as limitations of the study).
Adding the information suggested in the theoretical framework, the comparison of these results with other studies could be expanded so that the focus is not so much on describing one's own results.
Author Response
Reviewer 1
The text must be marked with line numbers. Was marked
This study is very brief and focuses mainly on descriptive statistical data. It would be more appropriate to explain which injuries and in which specific movements they occur. Lesions associated with the inguinal area and hip adductor musculature were specified, and their respective characterization.
There are parts of the introduction that should not be in this section (from the 2nd paragraph onwards). The concordance of the direction of asymmetry was also evaluated, so these paragraphs support the importance of asymmetry analysis.
More citations are needed on the use of platforms for this purpose and, if not available, look for sports with similar physiological demands. Examples of specific isometric and dynamic strength tests involving force platforms were mentioned and added.
Establishing as a requirement for inclusion that they can perform the test is an important bias. Details such as experience, previous injuries, etc., which are not mentioned, should be taken into account. Inclusion criterions were corrected.
The intensities of the warm-up are not quantitatively indicated and this is an important aspect for the main part. Added.
It would be convenient to indicate why you have chosen a static action and not a dynamic one (with a dynamometer for this action). It will then be easier to perform it in a standing position, as your sport is developed. The choice of an isometric action as the focus of this study responds to the main objective of evaluating the reliability of a specific test of adductor strength, analyzing its intra- and inter-test consistency. Although eccentric contractions can be quantified using inertial devices, their inclusion would modify the methodological approach of the study and would not be directly comparable to the specificity of isometrics in terms of neuromuscular demand.
Furthermore, the use of force plates in this assessment allows for accurate and objective data on the peak force generated, ensuring greater validity in the quantification of isometric force. This reinforces the practical applicability of the test in sports environments, where rapid and accurate measurement is key to decision making in load management and physical preparation.
It could also be suggested to perform an eccentric test, which could also be quantified with inertial devices (at least these two considerations could be considered as limitations of the study).
Añadiendo la información sugerida en el marco teórico, se podría ampliar la comparación de estos resultados con otros estudios para que el foco no esté tanto en describir los propios resultados. To point out that, although eccentric measurement can provide complementary information, the theoretical and methodological framework of this study is focused on validating the reliability of the isometric test, which is applicable and relevant in the evaluated context.
Finally, comparison with eccentric tests could introduce confusion in the interpretation of the results, as these assessments are not aligned with the parameters measured in this case. Therefore, we recommend keeping the focus on isometric assessment as the basis of the analysis to avoid unnecessary deviations from the main objective of the study.

Reviewer 2 Report
Comments and Suggestions for Authors
Many thanks to the Editor for the opportunity to revise the following non-experimental, descriptive study, “Intra and inter-test reliability of isometric hip adduction strength test with force plates in professional soccer players”, in which the Authors have investigated the reliability of a new isometric hip adduction test in a sports population.
Overall, the work is well conceptualised and written; however, I would like some modifications and/or clarifications, as follows:
Introduction, line 9. It is missing a space between “shown that the lower limbs” at the start of the referred line.
Heading 2.4.2. At the very last word, it is missing a parenthesis, closing the term “Figure 1”.
Tables 1 and 2. In the legend of Tables 1 and 2 is written “IL Lower limb 95%”. Would you mean ”Lower Limit 95%?”
2.4. May I ask you to insert the three sections of the warm-up procedures without bullet points?
2.4.2. Isometric Knee Extension recording. I think that it is missing the Figure of this test, as it is referring to Figure 1; however, Fig. 1 represents the isometric strength test. If I’m mistaken, I apologise; however, it needs more clarification.
In addition, can You insert some background information in the introduction section about the usefulness of Isometric Knee Extension?
2.5.1. Countermovement Jump processing. Why do you cite a Countermovement jump test (CMJ)?
Please can You expand the introduction section to insert the rationale behind the CMJ and the asymmetry concept?
May I ask you to insert in the 2.1 Design section all the tests/procedures (in the correct order) that You have assessed? With the recovery time between them.
Finally, in a practical approach, how much time is requested to perform the isometric hip adductor test in a 23 player’s team? How much time is requested to elaborate on all the results and give appropriate feedback to the medical and technical/tactical staff? Would it be applicable to implement this test soon after a warm-up but before training to assess the readiness to play and manage the daily workload? Can You expand on this thought?
Author Response
Reviewer 2
Introduction, line 9. A space is missing between “demonstrated that the lower limbs” at the beginning of the line referred to. Corrected
Title 2.4.2. In the last word, a parenthesis is missing to close the term “Figure 1”. Corrected
Tablas 1 y 2. En la leyenda de las Tablas 1 y 2 se escribe “IL Miembro Inferior 95%”. ¿Querrá decir “Límite Inferior 95%?” Corrected
2.4 . May I ask you to insert the three sections of the warm-up procedures without bullets? Corrected
2.4.2. Isometric Knee Extension Record. I believe the Figure for this test is missing, as it refers to Figure 1; however, Figure 1 represents the isometric strength test. If I am wrong, I apologize; however, it needs further clarification. It was an error in transferring the manuscript to the journal format. The test was not part of the study.
Also, can you insert some background information in the introduction section on the usefulness of isometric knee extension? It was a mistake, the test was not part of the study.
2.5.1. Processing of countermovement jumps: Why is a countermovement jump test (CMJ) cited? It was a mistake, the test was not part of the study.
Could you expand the introduction section to insert the rationale behind the CMJ and the concept of asymmetry? It was a mistake, the test was not part of the study.
May I ask you to insert in section 2.1 Design all the tests/procedures (in the correct order) that you have evaluated? With the recovery time between them. We believe that what is requested is explained in the first paragraph of section “2.3 Procedure”.
“The players participated in an initial familiarization session. The evaluation was carried out the following day, corresponding to the second day of a preseason micro-cycle, and the second evaluation was carried out 7 days later. All evaluations were carried out at the club's facilities, in the morning and before the technical-tactical practice.”
Finally, in a practical approach, how much time is required to perform the isometric hip adductor test on a team of 23 players? How much time is required to elaborate all the results and give the appropriate feedback to the medical and technical/tactical staff? Would it be applicable to implement this test shortly after a warm-up but before training to assess readiness to play and manage the daily workload? Can you expand on this idea? The implementation of the isometric hip adductor test in a team of 23 players can be adjusted according to the available resources and the needs of the technical and medical staff. Considering two evaluators and the use of two force plates, the entire process, including warm-up, can be performed in approximately 45 minutes. This time allows for an efficient evaluation and minimizes interference in the players' routine.
As for data processing, with the standard software available, it takes approximately 4 to 5 hours to analyze the results and generate a detailed report, which could be ready the next day. However, by using premium licenses of the software, it is possible to obtain key data such as peak force instantaneously, allowing for immediate feedback.
This methodology is feasible to implement just after the warm-up and before the technical-tactical training, as it assesses the physical preparation of the players and facilitates a proper management of the daily workload. In fact, in our experience, players have been able to train without any inconvenience one hour after the evaluation, which shows the practicality of the test in the context of a sports routine.
It is added in practical applications of the text

Round 2
Reviewer 2 Report
Comments and Suggestions for Authors
I thank the Authors for responding and resolving each point.